

# Genome skimming is a low-cost and robust strategy to assemble complete mitochondrial genomes from ethanol preserved specimens in biodiversity studies

Bruna Trevisan[1,*], Daniel M.C. Alcantara[1,*], Denis Jacob Machado[1,2], Fernando P.L. Marques[1] and Daniel J.G. Lahr[1]

[1] Department of Zoology, Institute of Biosciences, University of São Paulo, São Paulo, São Paulo, Brazil

[2] Department of Bioinformatics and Genomics / College of Computing and Informatics, University of North Carolina at Charlotte, Charlotte, NC, United States of America

[*] These authors contributed equally to this work.

## ABSTRACT

Global loss of biodiversity is an ongoing process that concerns both local and global authorities. Studies of biodiversity mainly involve traditional methods using morphological characters and molecular protocols. However, conventional methods are a time consuming and resource demanding task. The development of high-throughput sequencing (HTS) techniques has reshaped the way we explore biodiversity and opened a path to new questions and novel empirical approaches. With the emergence of HTS, sequencing the complete mitochondrial genome became more accessible, and the number of genome sequences published has increased exponentially during the last decades. Despite the current state of knowledge about the potential of mitogenomics in phylogenetics, this is still a relatively under-explored area for a multitude of taxonomic groups, especially for those without commercial relevance, non-models organisms and with preserved DNA. Here we take the first step to assemble and annotate the genomes from HTS data using a new protocol of genome skimming which will offer an opportunity to extend the field of mitogenomics to under-studied organisms. We extracted genomic DNA from specimens preserved in ethanol. We used Nextera XT DNA to prepare indexed paired-end libraries since it is a powerful tool for working with diverse samples, requiring a low amount of input DNA. We sequenced the samples in two different Illumina platform (MiSeq or NextSeq 550). We trimmed raw reads, filtered and had their quality tested accordingly. We performed the assembly using a baiting and iterative mapping strategy, and the annotated the putative mitochondrion through a semi-automatic procedure. We applied the contiguity index to access the completeness of each new mitogenome. Our results reveal the efficiency of the proposed method to recover the whole mitogenomes of preserved DNA from non-model organisms even if there are gene rearrangement in the specimens. Our findings suggest the potential of combining the adequate platform and library to the genome skimming as an innovative approach, which opens a new range of possibilities of its use to obtain molecular data from organisms with different levels of preservation.

Corresponding author
Daniel J.G. Lahr, dlahr@ib.usp.br

## INTRODUCTION

Global loss of biodiversity is an ongoing process that concerns both local and global authorities (*Velend, 2017*). Biodiversity loss impacts ecosystem functions, while additionally increasing the knowledge gaps and sampling biases (*Asaad et al., 2017*; *Oliveira et al., 2017*). Challenges including habitat loss, overexploitation, climate change, and invasive species are far from a solution (*Corlett, 2017*; *Velend, 2017*). Some advocate the use of natural history collections as a central tool for the study of biodiversity, especially for species that are becoming extinct or increasingly rare (*Staats et al., 2013*; *Kanda et al., 2015*). Studies of biodiversity mainly involve traditional methods using morphological characters and molecular protocols predominantly by PCR-based methods. However, conventional techniques are a time consuming and resource demanding tasks (*Cameron, 2014a*; *Yuan et al., 2016*). Among molecular approaches, the PCR-based methods are often not successful in recovering genetic data of preserved organisms, due to the fragmented nature of old and poorly-preserved DNA (*Heintzman et al., 2014*; *Timmermans et al., 2016*). Also, the lack of genomic resources such as well-established and optimized molecular markers and primers to delimit target amplicons for closely related species may hamper PCR-based methods for non-model organisms (*Ekblom & Galindo, 2011*; *Tilak et al., 2015*; *Matos-Maraví et al., 2019*).

The development of high-throughput sequencing (HTS) techniques has reshaped the way we explore biodiversity and opened a path to new questions and novel empirical approaches (*Dodsworth, 2015*; *Linard et al., 2015*; *Porter & Hajibabaei, 2018*). The use of low-coverage and cost-effective genome-skimming, also known as whole-genome shotgun sequencing (WGS), is one of these techniques. This particular method consists of sequencing the whole genome of an individual at low nuclear genome coverage. The process provides an extensive data set, capable of recovering high-copy fractions of total genomic DNA (organellar genomes, nuclear ribosomal DNA, and other multi-copy elements) through random shearing and inexpensive multiplexing (*Berger et al., 2017*; *Matos-Maraví et al., 2019*). The technique is potentially efficient for old museum material and ethanol-preserved specimens (*Staats et al., 2013*; *Maddison & Cooper, 2014*; *Linard et al., 2016*; *Grandjean et al., 2017*). Despite recent efforts to obtain DNA sequences through HTS protocols from museum specimens (*McCormack et al., 2017*; *Dabney et al., 2013*; *McCormack, Tsai & Faircloth, 2016*) and ethanol material (*Raposo do Amaral et al., 2015*; *Brabec et al., 2015*; *Brabec et al., 2016*; *Hartikainen et al., 2016*; *Vanhove et al., 2018*), the potential use of genome skimming for this purpose remains unexplored. There is a variety of suitable HTS sequencing platforms and library options to choose from to use in combination with genome skimming, including PCR-based libraries and PCR-free libraries that are less-error prone but require higher input DNA ($\leq$ one μg) (*Twyford & Ness, 2017*). Knowing that the preservation level of biological samples can influence the

quality of sequencing, it is important to consider the amount of input DNA available. Since researchers often preserve specimens of invertebrates in ethanol, specific protocols are required to obtain high-quality data from low quality or quantity DNA extracts (*Tilak et al., 2015*; *Linard et al., 2016*). Therefore, combining the adequate platform and library to the genome skimming technique is an innovative approach, which could overcome most of the limitations highlighted above, opening a new range of possibilities of its use for obtaining molecular data.

The mitogenome has been used as a molecular marker in a great variety of studies (e.g., ecology, evolution, phylogeography and phylogenetics at multiple taxonomic levels; see *Avise et al., 1987*; *Le et al., 2000*; *Zarowiecki, Huyse & Littlewood, 2007*; *Avise, 2012*; *Li et al., 2017*). Its popular use throughout those areas could be attributed to its particularity as maternal inheritance, high copy-number, lack of recombination and higher mutation rate when compared to other markers (*Ballard & Whitlock, 2004*; *Hahn, Bachmann & Chevreux, 2013*; *Yuan et al., 2016*; *Li et al., 2017*). Animal mitochondrial genomes are generally uniform across metazoan groups (*Park et al., 2006*; *Tan et al., 2017*): a circular, double-stranded DNA molecule, ranging from 15–20 kb in size, containing circa 37 genes (i.e., two mitochondrially encoded ribosomal RNAs [rDNA], 13 protein-coding genes [PCG] and 22 transfer RNA genes [tRNA]) (*Park et al., 2006*; *Castellana, Vicario & Saccone, 2011*; *Bernt et al., 2013*; *Tan et al., 2017*). This set of attributes provide to mitogenomes a wide spectrum of informational content, which can be used to answer many biodiversity questions.

Despite the potential of mitogenomes in solving biodiversity questions, the great majority of the studies only targeted a small fraction of this genome. The prevalence of partial sequences of the mitochondrially encoded ribosomal RNAs MT-RNR1 and MT-RNR2 (12S and 16S, respectively), Cytochrome B (MT-CYB) and Cytochrome C Oxidase I (MT-CO1) in many studies can be credited to the existence of "universal primers" that amplified these regions for a whole spectrum of non-model Metazoa taxa. Hence, it is not uncommon to find studies addressing phylogenetic relationships at different levels of divergence—including the interest on detecting cryptic species—based on these markers (*Von Nickisch-Rosenegk, Lucius & Loos-Frank, 1999*; *Zehnder & Mariaux, 1999*; *Hu et al., 2005*; *Wickström et al., 2005*; *Littlewood, Waeschenbach & Nikolov, 2008*; *Brabec et al., 2016* and *Vanhove et al., 2018* to cite a few). However, to date HTS provided the means of sequencing the complete mitochondrial genome in reasonable time and at relative low cost. As a result, the number of genome sequences published has increased exponentially during the last decades (*Park et al., 2006*; *Hahn, Bachmann & Chevreux, 2013*; *Tan et al., 2017*; *Raposo do Amaral et al., 2015*; *Vanhove et al., 2018*). With the increase of studies using complete mitogenomes, several authors have recognized the virtues of a greater amount of nucleotide sequence data for inferring robust phylogenies in many taxonomic groups such as mammals (*Arnason et al., 2002*; *Campbell & Lapointe, 2011*), birds (*Pacheco et al., 2011*), insects (*Cameron, 2014b*), and flatworms (*Brabec et al., 2015*; *Brabec et al., 2016*; *Maldonado et al., 2017*; *Vanhove et al., 2018*). In addition, we can also relate the power of resolution of the mitogenome to its genome-level characteristics such as gene arrangements and the positions of mobile genetic elements, which are good alternatives
to resolve deeper phylogenetic questions (*Waeschenbach, Webster & Littlewood, 2012*; *Guo, 2015*; *Li et al., 2017*).

In spite of its undeniable informational content, whole mitogenomes are still relatively under explored in phylogenetic studies for a multitude of taxonomic groups, especially for those without commercial relevance, non-model organisms, and preserved DNA (*Littlewood, Waeschenbach & Nikolov, 2008*; *Maldonado et al., 2017*). The majority of these groups still have poorly understood phylogenetic histories. Two examples of such groups are cestodes in the family Rhinebothriidae and dipterans of the family Streblidae. Rhinebothriideans are exclusively endoparasites of batoid elasmobranchs, while Streblids are highly specialized ectoparasites of bats (*Dick, Graciolli & Guerrero, 2016*; *Ruhnke, Reyda & Marques, 2017*). Both groups have intricate historical associations with their hosts; which are of scientific interest of evolutionary biologists engaged in understanding how historical association events shaped the ecology, patterns of association and evolution of these host/parasite systems (*Wenzel, Tipton & Kiewlicz, 1966*; *Brooks, Thorson & Mayes, 1981b*; *Dick & Patterson, 2006*; *Tello, Stevens & Dick, 2008*; *Marques & Caira, 2016*).

However, despite the efforts to reconstruct the phylogenies using fragments of mtDNA (or, in the case of rhinebothriideans, only pieces of rRNA genes), their internal relationships remain poorly understood. We could attribute this lack of understanding to a number of factors including the difficulty in extracting DNA from fixed organisms, the low resolution of those markers, and the limited availability of sequenced samples (*Dittmar et al., 2006*; *Petersen et al., 2007*; *Caira et al., 2014*; *Ruhnke, Caira & Cox, 2015*; *Trevisan, Primon & Marques, 2017*). Thus, mitogenomics carries the potential to resolve the phylogenetic history in those groups. Here we take the first step to assemble and annotate the genomes from HTS data using a new genome skimming protocol, revealing an opportunity to extend the field of mitogenomics to under-studied organisms.

## METHODS

### Taxon sampling

We fixed all samples in 96% ethanol and stored them at –20 °C until we performed the genomic DNA extractions. Hologenophores (sensu *Pleijel et al., 2008*) from Rhinebothriidae were deposited at MZUSP (Museu de Zoologia da Universidade de São Paulo, Universidade de São Paulo, São Paulo, SP, Brazil). We collected the batflies in Brazil, *Paradyschiria parvula* Falcoz, 1931 from Base de Estudos do Pantanal, Passo do Lontra, Corumbá, Mato Grosso do Sul (19°34′48.0″S, 57°01′15.1″W) in 2013 and *Paratrichobius longicrus* (Miranda Ribeiro, 1907) from Núcleo Pedra Grande, Parque Estadual Cantareira, São Paulo, São Paulo (23°26′10.9″S, 46°38′07.8″W) in 2017. We collected the hosts following the permit guidelines issued by Sistema de Autorização e Informação em Biodiversidade - SISBIO (5184-1, issued in 2013, Brazil to Gustavo Graciolli from Universidade Federal do Mato grosso do Sul to sample *Paradyschiria parvula* and by SISBIO and by Secretaria do meio Ambiente - SMA to sample *Paratrichobius longicrus* (55242-1 and 260108–008.107/2016, respectively) both issued in 2016, Brazil to Daniel Máximo Corrêa de Alcantara. Under those permits, we captured these hosts using mist

 

nets, opened at ground level in trails and other locations near to bodies of water during 6 h after sunset, and checked them every 30 min. We examined the bats for ectoparasites manually or with the aid of tweezers. Bat collection procedures were approved by Comissão de Ética no Uso de Animais, Instituto de Biociências, USP (Proc. 16.1.448.41.6).

We obtained *Anindobothrium anacolum* Marques, Brooks & Lasso, 2001 (Voucher MZUSP 7968) and *Rhinebothrium reydai* Trevisan & Marques, 2017 (Voucher MZUSP 7969) from the spiral intestines of the stingray *Styracura schmardae* from Trinidad & Tobago (Maracas, San Juan-Laventille, 10°45′N, 61°26′W) in 2014 and from Panama (Almirante, Bocas del Toro, 9°17′N, 82°20′W) in 2015, respectively. We collected these hosts using spears following the permit guidelines issued by the Ministry of Food Production—Fisheries Division (issued in September 2014, Trinidad & Tobago to F.P.L. Marques) and by the Autoridad Nacional del Ambiente—ANAM (SE/A-101-14, issued in December 2014, Panama to F.P.L. Marques), respectively. Further details on the collection of hosts and specimens preparation is available in *Trevisan, Primon & Marques (2017)* and *Marques & Reyda (2015)*.

## DNA extraction

For Streblidae, we extracted DNA using the Qiagen DNeasy Blood & Tissue Kit (Qiagen). Since it is common the abdomen of Streblidae contain the host's blood, we separated the thorax of each specimen from the abdomen, and only the thorax with the head and legs were used to avoid contamination. The DNA was eluted in 200 µl of a buffer solution, repeating the elution step twice with the addition of 100 µl each time for increased DNA yield. After extraction, we stored the thorax with the head and legs in ethanol, together with the specimen abdomen, and cataloged as specimen vouchers.

For Rhinebothriidae, we extracted the DNA from the middle portion of the strobila of each specimen, which was removed and allowed to air dry for about 5 min at room temperature. We extracted total genomic DNA using Agencourt DNAdvance—Nucleic Acid Isolation Kit (Beckman Coulter, Brea, CA, USA) following the manufacturer's instructions. We prepared scolices and posterior portions of strobila from specimens used in molecular analyses as whole mounts following traditional protocols (*Trevisan, Primon & Marques, 2017*).

We employed standard precautions to minimize contamination throughout, such as using exclusive pipettes with filter tips and bleaching all the instruments used in DNA extraction. We measured the purity and amount of DNA extractions using a NanoDrop 2000 spectrophotometer (Thermo Fisher Scientific, Waltham, MA, USA) and Qubit 2.0 Fluorometer using Qubit high sensitivity dsDNA assays (Life Technologies, Carlsbad, CA, USA).

## Library preparation and sequencing

We used Nextera XT DNA Library Preparation Kit (Illumina) to prepare indexed paired-end (PE) libraries according to the manufacturer's protocol. We chose Nextera XT to prepare the libraries because the fabricator optimized the protocol for one ng ($5 \times 0.2$ ng/µl) of input DNA in total. The low amount of input DNA provides a powerful tool for working

with a variety of samples that yield either small or copious amounts of tissue. This library is also likely to be suitable for DNA extractions from samples of model and non-model taxa with different ages of fixation, especially for small genomes ($\leq$5 Mb), PCR amplicons, and plasmids.

Before starting libraries preparation, we diluted DNA extracts in Milli-Q water to 0.2 ng/µl, after which we checked the concentration in a Qubit 2.0 Fluorometer. We used a low-cost method to determine the quality and size of the sequencing libraries, as performed in *Kang et al. (2017)*. The technique consists of PCR amplification of the library, using Illumina adapter primers, checking amplicons for quality and size by standard agarose gel electrophoresis. We prepared a PCR master mix with the appropriate volume for each sample, containing 5 µl of KAPA Taq ReadyMix PCR Kit, 3 µl of Milli-Q water, 0.5 µl of Illumina forward primer (10 µM), 0.5 µl of Illumina reverse primer (10 µM) and 1 µl of DNA template. Then, we run PCR using the following protocol: 45 s of initial denaturation at 98 °C; 20 cycles of 25 s of denaturation at 98 °C, 30 s of annealing at 47 °C, and 1 min and 30 s of extension at 72 °C; 3 min of final extension at 72 °C; and hold at 4 °C. Subsequently, we examined these PCR products in 1.8x TBE agarose gel electrophoresis. We determined the sequencing library concentrations on Qubit 2.0 Fluorometer using Qubit high sensitivity dsDNA. Library normalization was done manually, diluting libraries to the same concentration (four nM) before volumetric pooling.

We sequenced the samples of Rhinebothriidae and Streblidae in two different Illumina platforms. Samples of Streblidae were sequenced alone using an Illumina MiSeq System, with a Reagent Kit v3 to generate PE reads of 300 bp. Since it is possible to sequence 24–30 million reads with the specifications used to run the Illumina MiSeq System, we pooled up to two DNA libraries. The samples of Rhinebothriidae were sequenced using an Illumina NextSeq 550 System, with a High-Output Kit to generate PE reads of 150 bp. This system can sequence up to 800 million reads with these specifications, allowing us to pool up to 35–40 DNA libraries. Thereby, each lane of the Illumina NextSeq 550 also received 33 additional libraries. We based the calculation of the number of reads required per sample to recover the complete mitochondrial genome on *Richter et al. (2015)*. We performed all DNA sequencing in the Core Facility for Scientific Research—University of São Paulo (USP) (CEFAP-USP).

## Computational resources

We executed *in silico* procedures using "ACE", an SGI rackable computer cluster housed in the Museum of Zoology of the University of São Paulo. Select servers had four 2.3 GHz Operon CPUs with 16 cores each and 256 or 516 GB of memory. The software environment in ACE consists of a SUSE Linux Enterprise Server with SGI Performance Suite, SGI Management Center and PBS Pro Job Scheduler. We were able to reconstruct each genome using a single core and ca. 20 GB of memory.

## Quality control and mitogenome assembly

We pre-processed the raw reads from each pair using a series of UNIX commands. We trimmed and filtered the sequences using the HTQC toolkit (*Yang et al., 2013*) a home-made Python script (selectTiles.py, see *Machado, Lyra & Grant 2016*) that automates tiles

selection. We evaluated the quality of filtered reads with FASTQC (*Andrew, 2010*). The assembly protocol received only filtered PE reads. We described the complete quality control protocol below and the step-by-step procedures are given in *Machado, Lyra & Grant* (*2016*, Appendix S1).

We performed the sequence assembly using a baiting and iterative mapping strategy based on MIRA v4.0 *Chevreux, Wetter & Suhai (1999)* and a modified version of MITOBIM.PL v1.6 (*Hahn, Bachmann & Chevreux, 2013*), following the guidelines described in *Machado, Lyra & Grant* (*2016*, Appendix S2). We applied the same search parameters to every assembly but used different baits depending on the class of the specimen. The reference mitogenome sequence of the house fly (*Musca domestica* L., GenBank Accession Number KM200723) was the bait for all streblids, and the reference mitogenome sequence of the beef tapeworm (*Taenia saginata Goeze, 1782*, GenBank Accession Number NC_009938) was the bait for the assembly of cestodes. Finally, we inferred the completeness of each new putative mitogenome (i.e., sequence circularization) using the AWA program and the contiguity index statistics described in *Machado et al., 2018*; the AWA beta version is available at https://gitlab.com/MachadoDJ/awa.

We mapped the raw sequence reads back to the putative mitogenome selected by AWA with Bowtie2 v2.2.6 (*Langmead & Salzberg, 2012*) using the local alignment algorithm and the highest sensitivity setting. We set the threshold for base calling on the consensus sequence to bases that match at least 99% of the sequences, with a minimum coverage per position of three sequences.

## Mitogenome annotation

We parsed the assemblies in CAF format using a home-made Python script (parseCaf.py; see *Machado, Lyra & Grant, 2016*) to extract DNA data and evaluate the coverage and quality of each mtDNA element. Preliminary *de novo* mitogenome annotation used the mitochondrial genome annotation server MITOS2 (*Bernt et al., 2013*, available at http://mitos2.bioinf.uni-leipzig.de), changing the genetic code accordingly (transl_table=9 for flatworm, transl_table=4 for insects).

We applied three different strategies independently to corroborate the annotations of coding genes. We used the BLAST (*Altschul et al., 1990*) to search a selected database of mitochondrial peptides from UniProt/Swiss-Prot (*The UniProt Consortium, 2016*; UniProt resources are available at https://www.uniprot.org/). We executed a second comparison between reference amino acid sequences and the new nucleotide sequences with GeneWise (*Birney, Clamp & Durbin, 2004*). Finally, we also applied TransDecoder (see *Hahn, Bachmann & Chevreux, 2013*; the program is available at https://github.com/TransDecoder) to identify candidate coding regions and compare the outputs from these programs to propose the final annotations.

We performed additional search and validation of tRNA sequences using ARWEN (*Laslett & Canbäck, 2007*) and tRNAscan-SE (*Lowe & Eddy, 1997*; *Schattner, Brooks & Lowe, 2005*). We confirmed and edited manually the automated annotation by comparison to published reference mitogenomes of flies and tapeworms. We annotated the control region (CR) with sequence similarity searches in BLAST using default parameters.

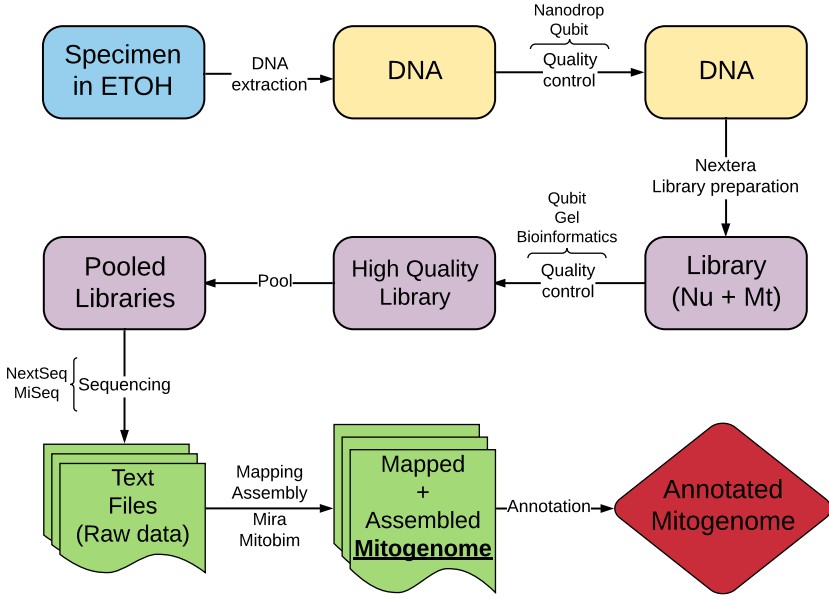

**Figure 1    Schematic workflow of the new protocol proposed in this study.**

Annotation of cestode mitogenomes was less straightforward compared to streblid organelles and required a more complex strategy. This was needed because some software mentioned above and used for annotation have not yet fully implemented the traditional codon table for flatworms (transl_table=9). Besides, nucleotide and amino acid reference sequences were not available. These limitations reduced the efficiency of the annotations of flatworm sequences, resulting in some wrong start or end positions or missing genes. Although they did not impede our semi-automatic annotation strategy, the procedure was very time-consuming and required manual curation. Figure 1 shows a summary of the workflow of the protocol we proposed in this study.

## RESULTS

### DNA extraction, library preparation and genome sequence

We obtained sufficient DNA quantities for all samples (Table 1), but we required some additional protocols to achieve the initial concentration for library preparation. We diluted the samples from *A. anacolum* and *R. reydai* because they contained a DNA concentration higher than initially required. The DNA concentration obtained for *P. longicrus* was closer to the amount necessary to prepare the library, and therefore its DNA was not diluted. In the case of *Paradyschiria parvula*, the DNA concentration was too low ($\leq 0.50$ ng/ml). In this particular case, we concentrated the extracted DNA using Agencourt AMPure XP and eluting the DNA in 15 µl. This protocol allowed us to reach the concentration close to that necessary to start the library preparation, without the need for dilution (Table 1). The sequence on Illumina MiSeq resulted in a total of 13.98 mi PE reads for *P. parvula* and 16.82 mi PE reads for *P. longicrus*. For Rhinebothriidae, the sequence on Illumina NextSeq resulted in a total of 11.81 mi PE reads for *A. anacolum* and 7.04 mi PE reads for *R. reydai*.

**Table 1 Concentration of DNA in the extraction and library preparation from the specimens included in this study.** The measurements were performed in Qubit 2.0 Fluorometer and the values are given in ng/µl. A, after DNA extraction; B, after using Agencourt AMPure XP; Dilution, after diluting the DNA extracted to start the library preparation; Final, after library preparation with Nextera XT.

| Species | Extraction | | Library Prep | |
|---|---|---|---|---|
| | **A** | **B** | **Dilution** | **Final** |
| *Paradyschiria parvula* | ≤0.1 | 0.184 | – | 2.12 |
| *Paratrichobius longicrus* | 0.354 | – | – | 3.34 |
| *Anindobothrium anacolum* | 18.6 | – | 0.208 | 14.2 |
| *Rhinebothrium reydai* | 9.08 | – | 0.260 | 12.9 |

## Mitogenome assembly

The present study is the first to report the complete mitochondrial genome of species from the two families, Streblidae and Rhinebothriidae. Figure 2 illustrates the annotation and gene map of the new mitogenomes. The complete mitogenome sequences obtained for the two species of Streblidae were 14,588 bp and 16,296 bp for *Paradyschiria parvula* (GenBank accession no. MK896865) and *Paratrichobius longicrus* (GenBank accession no. MK896866), respectively. The average coverage for the assemblies of Streblidae was high, 1,749.6 for *Paradyschiria parvula* and 6,355 for *Paratrichobius longicrus*. Their mitogenome length conforms with those found in other Diptera, typically 14–19 kb (*Li et al., 2015*). The total number of mapped sequences for *Paradyschiria parvula* was 96,404 whereas *Paratrichobius longicrus* shown a higher value of 695,060. However, circularity tests of the mitogenome of *Paradyschiria parvula* presented a higher average coverage, contiguity, and score than *Paratrichobius longicrus* (329 vs. 37.3, 322.3 vs. 36.3 and 1.43 vs. 6.79, respectively) (Table 2). The lower score of *Paratrichobius longicrus* is bound to be affected by the ambiguous nucleotides obtained from our conservative approach to base calling (detailed below).

The mitogenome sequences obtained for specimens of Rhinebothriidae, *A. anacolum* (GenBank accession no. MK887326) and *R. reydai* (GenBank accession no. MK896864), were 13,693 bp and 13,506 bp in length with 743.4 and 258.5 coverage, respectively. Their mitogenome size follows the pattern observed previously for tapeworms (13–15 kb) (*Li et al., 2017*). The total number of mapped sequences for *A. anacolum* was 83,969 and *R. reydai* shown a smaller value of 28,583. We observed the opposite for the contiguity index in which *A. anacolum* had a lower value in comparison to *R. reydai* (153.2 vs. 313.8, respectively). The GC content and quality of the sequences was similar for both species (30.4% vs. 35.8% and 34.0 vs. 32.4, respectively). Overall the scores *A. anacolum* and *R. reydai* were close to 0, which should be considered as another good indicator of quality (non-ambiguity) (Table 2).

## Mitogenome organization and structure
### Streblidae

The mitogenome of both species contains 37 genes, including 13 PCGs, 22 tRNA genes, two rRNA genes and the CR. We recovered 71 bp of the CR for *P. parvula* and 1,579 bp for *P. longicrus*. For both species, 23 genes are encoded on the majority strand (14 tRNAs

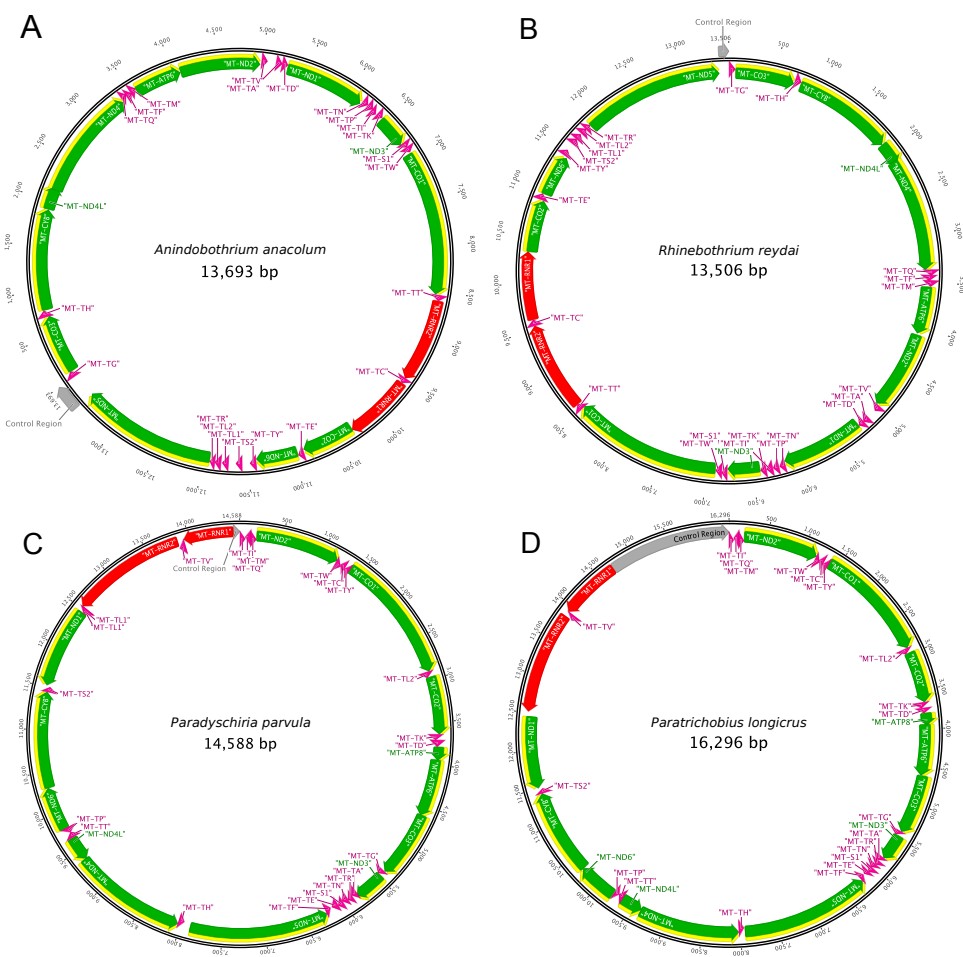

**Figure 2** Graphical representation of the mitogenomes of **(A)** *Anindobothrium anacolum*, **(B)** *Rhinebothrium reydai*, **(C)** *Paradyschiria parvula*, **and (D)** *Paratrichobius longicrus*. Grey: control region; yellow: CDS; green: genes; red: rRNA; pink: tRNA.

**Table 2  General assemble statistics.** Q1 and Q3 indicate 1st and 3rd quartiles for coverage, respectively. Asterisks indicate indexes calculated during circularization tests in AWA (*Machado et al., 2018*) using the 50 nucleotides flanking each end of the putative mitochondrion contig.

| Species | Length | GC% | No. of sequences | Avg. Coverage (Q1 : Q3) | AWA | | | |
|---------|--------|-----|------------------|-------------------------|-----|---|---|---|
| | | | | | Avg. Coverage* | Avg. Contiguity* | Avg. Quality* | Bowtie2 Score* |
| *P. parvula* | 14,588 | 21.4 | 96,404 | 1,749.6 (841 : 2,133) | 329 | 322.3 | 35.8 | −1.43 |
| *P. longicrus* | 16,296 | 17.9 | 695,060 | 6,355.0 (5,506 : 7,945) | 37.3 | 36.3 | 36.0 | −6.79 |
| *A. anacolum* | 13,693 | 30.4 | 83,969 | 743.4 (526 : 962) | 157.3 | 153.2 | 34.0 | −0.92 |
| *R. reydai* | 13,506 | 35.8 | 28,583 | 258.5 (216 : 305.3) | 318.9 | 313.8 | 32.4 | −2.45 |

and nine PCGs), while the minority strand encodes the remaining 14 genes (eight tRNAs, the two ribosomal RNAs, and four PCGs) (Fig. 2). We must note a bias in the nucleotide composition of the mitogenome toward A and T, with a GC content of 21.4% for *P. parvula* and 17.9% for *P. longicrus* (Table 2). Except for MT-CO1, each of the thirteen PCGs had the canonical start codon ATT (encoding Ile) or ATG (encoding Met). We identified the MT-CO1 start codon as TCG (encoding Ser). We found incomplete stop codons (T) only in MT-ND4 for *P. parvula*. All other PCGs have the complete stop codons TAA or TAG in both species, being TAG found only for MT-CYB. We also found several stop codons TGA within all PCGs.

### Rhinebothriidae

Each mitogenome contains 36 genes, including 12 protein-coding genes (MT-ATP6, MT-CO1-3, MT-CYB, MT-ND1-6 and MT-ND4L), 22 transfer RNA genes (tRNA), two ribosomal RNA genes (RNR1-2) and one CR. As previously reported for other Neodermata mitogenomes, the specimens from Rhinebothriidae also lacks the ATP8 gene, which is found in other metazoan mitogenomes (*Le, Blair & McManus, 2002*; *Guo, 2016*; *Zhao et al., 2016*; *Li et al., 2017*; *Egger, Bachmann & Fromm, 2017*). We noticed that the nucleotide composition of the mitogenome is biased towards A and T, with a GC content of 35.8% for *R. reydai* and 30.4% for *A. anacolum* (Table 2). All genes are encoded in the same strand and transcribe from the same direction. The gene order in *A. anacolum* and *R. reydai* follows the typical organization of cestodes, except by the rearrangements of some tRNA genes and by the number of non-coding regions (NCR) (i.e., one). Despite the slightly differences, these results reinforces previous evidences that cestode gene order is extremely conserved (*Nakao, Sako & Ito, 2003*; *Li et al., 2017*; *Zhang et al., 2017*). The reader should note that annotations described here for the mitogenomes of flatworms derive from a semi-automatic and manually curated annotation protocol but, given the specific challenges involved in annotating this mitogenomes, some of the authors from this publication prepared a pipeline dedicated to their annotation, which is available at https://gitlab.com/MachadoDJ/cma.

## DISCUSSION

### Coverage variation

The overall quality and alignment score of the assembly of the four mitogenomes indicate that even with relatively lower coverage values, it is possible to recover the mitochondrial genome with this protocol. Given our conservative approach towards base calling, we left some ambiguous nucleotides in six tRNAs, two rRNAs, 11 PCGs and the CR of the mitogenome of *P. longicrus*. However, the number of these ambiguities is within our expectations given the procedures described here and the expected variation in the percentage of mitochondrial among different libraries.

As expected, there was a wide variation on the number of reads mapped to each mitochondrial genome. The ratio between the number of mitochondria and the genome size of the organism influences the number of mitochondrial reads in a library (*Richter et al., 2015*). In tunicates, for instance, a lower proportion of mitochondria to nuclei in tissues is

correlated to reduced numbers of mitochondrial reads (*Tilak et al., 2015*). The mitogenome coverage may even correlate negatively with the amount of DNA used for sequencing, since higher amounts of DNA can increase the chance of introducing more nuclear reads into the library, reducing the number of mitochondrial reads. Furthermore, the number of multiplexed species per sequencing can also influence the number of mitochondrial reads sequenced and the average coverage (*Richter et al., 2015*; *Tilak et al., 2015*). Therefore, the number of recovered mitochondrial reads in different genome-skimming experiments can vary.

The quality of the assembly around the 50 bp flanking each end of the putative mitogenomes is also presumed to vary. For the assembly of *P. longicrus* mitogenome, which has the lowest coverage and contiguity score in AWA, our analysis found *k*-mer lengths that obtained an average coverage higher than that presented in Table 2 (up to 207.73x). However, the identity and quality of these alignments were much lower, and therefore they were not selected by AWA software based on the combined criterion of read coverage, read contiguity, average quality, and alignment scores.

## Streblidae mitogenome order rearrangements

Compared to other dipterans, the gene order and sizes follows the typical organization reported for this group (*Nelson et al., 2012*; *Cameron, 2014b*; *Li et al., 2015*; *Pu et al., 2017*). Among flies, ATG (Met) and ATT (Ile) are frequently found as start codons (*Li et al., 2015*), which is congruent with our results. On the other hand, start codons for MT-CO1 are usually considered non-canonical in holometabolans, though TCG is widely reported for MT-CO1 in Diptera (*Cameron et al., 2007*; *Nelson et al., 2012*). The stop codon most commonly found in Diptera is TAA. However, partial stop codon T has been reported in many insect mitogenomes and is completed to a full TAA stop codon via post-transcriptional polyadenylation (*Li et al., 2015*). Moreover, some authors have already reported that translation termination might reassign the codon UGA to code for tryptophan or cysteine (*Alkalaeva & Mikhailova, 2017*), which may be the case in this study. Even though the CR can be variable in both size and nucleotide composition, the difference in size between the two species of Streblidae is striking. More study are important to investigate such difference, since cases of duplication in the CR are known in different groups, including insects (*Cameron, 2014b*; *Yan et al., 2012*).

Although Diptera is one of the most extensively sequenced orders within Insecta, the number of complete mitogenomes is low given its diversity (*Liu et al., 2017*; *Narayanan Kutty et al., 2019*). Streblidae belongs to the Calyptratae clade, one of the most species-rich group within Diptera with over 22,000 species. Calyptratae is divided into three superfamilies: Hippoboscoidea (to which Streblidae belongs), Muscoidea and Oestroidea (*Narayanan Kutty et al., 2019*). Currently, 231 mitochondrial genome sequences of Diptera are included in the NCBI GenBank Organelle Genome Resources (see also *Benson et al., 2008*). However, 51 of these mitochondrial genomes are from Calyptrates, while 41 are from Oestroidea, nine from Muscoidea and only one from Hippoboscoidea. It is clear that the Calyptratae clade is poorly represented, and the two new sequences will allow a significant gain not only for the family in question but also at many levels in insect systematics.

## Cestode mitogenome order rearrangements

The most recent and comprehensive study on the diversity of mitogenomes in cestodes was published by *Li et al. (2017)*. In this study, the authors included 54 mitogenomes representing 5 of the 18 orders presently recognized for cestodes (following *Caira et al., 2014*). The study included members of Caryophyllidea, Diphyllobothriidea, Bothriocephalidea, Onchoproteocephalidea and Cyclophyllidea; however, the latter order comprised 75% (41) of the species included in the study. Hence, this is the first attempt to document the diversity of mitogenomes within this group.

The authors considered that mitogenome gene order is extremely conserved in cestodes. Their assertion was based on the observation that all mitogenomes studied could be attributed to 4 categories based on the arrangements coding regions and tRNAs. According to *Li et al. (2017)*, caryophyllideans possesses a Category I mitogenome from which Category II derived by a transposition event in which the region $tRNA^{Leu(CUN)} - tRNA^{Ser(UCN)} - tRNA^{Leu(UUR)}$ translocated to the 3′ end of the four genes $cox2 - tRNA^{Glu} - Nad6 - tRNA^{Tyr}$. Following, two other categories would have derived from Category II; one by a tandem duplication and random loss (TDRL) event that generated the region $tRNA^{Leu(CUN)} - tRNA^{Leu(UUR)} - tRNA^{Tyr} - tRNA^{Ser(UCN)}$—Category III; and the other by a simple transposition event generating the region $tRNA^{Ser(UCN)} - tRNA^{Leu(CUN)}$—Category IV.

Although *Li et al.*'s (*2017*) study should be considered a preliminary account of the diversity of mitogenomes within cestodes, there appears to be some phylogenetic congruence among the patterns of gene rearrangements that can generate some predictions for future studies. The transposition that characterizes their Category II was found in all polyzooic orders included in the study (i.e., Diphyllobothriidea, Bothriocephalidea, Onchoproteocephalidea and Cyclophyllidea). The hypothesized TDRL resulting in the region $tRNA^{Leu(CUN)} - tRNA^{Leu(UUR)} - tRNA^{Tyr} - tRNA^{Ser(UCN)}$ found in three species of *Schyzocotyle* could be a putative synapomorphy for the genus, if not for the order Bothriocephalidea. Finally, the transposition leading to the region $tRNA^{Ser(UCN)} - tRNA^{Leu(CUN)}$ could be a synapomorphy for acetabutate cestodes (sensu *Caira et al., 2014*)—although it seems to be reverted to the ancestral state ($tRNA^{Leu(CUN)} - tRNA^{Ser(UCN)}$) in taeniids.

Our results confirm some of these predictions. Rhinebothriideans are considered to have derived after bothriocephalideans and the order is considered to be sister to a large clade comprised by six orders including Onchoproteocephalidea and Cyclophyllidea (see *Caira et al., 2014*). As predicted the mitogenomes of *Anindobothrium anacolum* and *Rhinebothrium reydai* share the region $tRNA^{Ser(UCN)} - tRNA^{Leu(CUN)}$ found in acetabutate cestodes, with the exception of teaniids (i.e., Category IV). However, contrary to most mitogenomes known to date for cestodes, our results indicate that rhinebothriideans have only one non-coding regions (NCRs), an attribute also found in the taeniid *Hydatigera taeniformis* (Batsch, 1786) (*Li et al., 2017*). However, taeniids seem to have reverted to the ancestral gene arrangement of Category II.

*Li et al. (2017)* acknowledged the possibility that the diversity of arrangements is underestimated given the taxonomic representativity of their dataset. In fact, when NCRs

are taken into account, we found that there are few arrangements that were not considered by the authors. For instance, the bothriocephalid *Schyzocotyle acheilognathi* posses a third NCR not found in any other cestode. Within Category IV, there are at least seven distinct arrangements if you consider the position of NCRs; *A. anacolum* and *R. reydai* yet have a different one. We predict that as we compile mitogenomes for cestodes, we will have a better understanding of the rearrangement events associated with the diversity of the group. To achieve this goal, it would be desirable to have representatives of all major lineages of cestodes and access variability in different taxonomic levels. We also think that special attention should be given to homology statements of NCRs since we already have an indication that the number of NCRs differ within the groups (see *Li et al., 2017*). Finally, we think that once we achieve the goals above and contextualize mitogenome rearrangements within a phylogenetic context we might uncover new synapomorphies for cestode taxa.

## Applicability of the method

Our results illustrated that the proposed protocol can successfully assemble mitochondrial sequences from genome skimming raw data of non-model organisms. We assembled the whole mitogenomes even if there were gene rearrangements, which is reinforced by the contiguity index supporting the circularization of those mitochondria.

The main advantage of this protocol is the possibility to start from a low concentration of DNA extracts (Table 1), circumventing the need for prior enrichment and can work well on samples with different levels of preservation. We believe that the critical point of this advantage lies in the library preparation kit used.

Many studies with genome skimming have used methods in which the user shears the genomic DNA through ultrasonication (*Kocher et al., 2014*; *Richter et al., 2015*) or the library prep kit requires a DNA input ≥ 50 ng (*Kocher et al., 2016*; *Linard et al., 2015*). Such methods would probably make it impossible to sequencing some of the samples used in this study (Table 1). For mitochondrial sequencing, Nextera XT is commonly used in conjunction with enrichment methods *via* organelle isolation (*Grandjean et al., 2017*) or PCR amplification (*Li et al., 2015*; *Foster et al., 2017*), but not with genome skimming techniques. Although the Nextera XT is designed for small genomes (≤5 Mb), such as genomes of bacteria and viruses, it has been used to recover mostly plastomes from plants (*Burke et al., 2016*).

Based on our results, we expect that the methods described here will be valuable to researcher aiming towards sequencing metazoan mitogenomes. The workflow is time-saving, and it is possible to go from DNA to the pool library in a single day. Moreover, researchers can certainly apply this protocol to other non-models organisms, in addition to old historical specimens or specimens that usually generate low concentrations of DNA from the extractions. We have demonstrated that mitochondrial genomes can be generated efficiently from different sequencing strategies, using Illumina MiSeq (two samples and PE reads of 300 bp) and Illumina NextSeq (35 samples and PE reads of 150 bp). Thus, the user can adjust the procedure costs by designing a multiplex pooling strategy that sequences the desired number of samples with suitable coverage.

## CONCLUSION

The proposed method is an excellent solution to obtain low cost/large scale molecular data in biodiversity studies. Combining the adequate platform and library to the genome skimming is an innovative approach, opening a new range of possibilities of its use in obtaining molecular data from organisms with different levels of preservation. The principal advantages from our approach are: (i) it requires low amount of input DNA (0.2 ng/μl), which allows the use of organisms with preserved DNA; (ii) it does not depends on specific primers and is not affected by gene rearrangement; and (iii) it is time-saving and cost-effective, leading to high-quality complete sequence assemblies.

## ACKNOWLEDGEMENTS

We thank the Core Facility for Scientific Research—University of São Paulo (CEFAP-USP/2017) and GENIAL (Genome Investigation and Analysis Laboratory)—for the sequencing on Illumina MiSeq and NextSeq and the Museu de Zoologia da Universidade de São Paulo (MZUSP) for granting the use of computational resources ("Ace" SGI computer cluster - FAPESP # 2012/10000-5).

### Funding

This work was supported by the Fundação de Amparo à Pesquisa do Estado de São Paulo, FAPESP grants #2013/04585-3, #2016/20792-7, #2017/11063-4 and #2018/03534-0. The funders had no role in study design, data collection and analysis, decision to publish, or preparation of the manuscript.

### Grant Disclosures

The following grant information was disclosed by the authors:
Fundação de Amparo à Pesquisa do Estado de São Paulo.
FAPESP: #2013/04585-3, #2016/20792-7, #2017/11063-4, #2018/03534-0.

### Competing Interests

Daniel J. Lahr is an Academic Editor for PeerJ.

### Author Contributions

- Bruna Trevisan and Daniel M.C. Alcantara conceived and designed the experiments, performed the experiments, analyzed the data, contributed reagents/materials/analysis tools, prepared figures and/or tables, authored or reviewed drafts of the paper, approved the final draft.
- Denis Jacob Machado analyzed the data, prepared figures and/or tables, authored or reviewed drafts of the paper, approved the final draft.
- Fernando P.L. Marques conceived and designed the experiments, analyzed the data, authored or reviewed drafts of the paper, approved the final draft.
- Daniel J.G. Lahr conceived and designed the experiments, analyzed the data, contributed reagents/materials/analysis tools, authored or reviewed drafts of the paper, approved the final draft.

## Animal Ethics

The following information was supplied relating to ethical approvals (i.e., approving body and any reference numbers):

Comissão de Ética no Uso de Animais, Instituto de Biociências, USP has provided full approval for this research (Proc. 16.1.448.41.6).

## Field Study Permissions

The following information was supplied relating to field study approvals (i.e., approving body and any reference numbers):

Bat hosts were collected following the permit guidelines issued by Sistema de Autorização e Informação em Biodiversidade - SISBIO (5184-1) and by Secretaria do meio Ambiente (SMA 55242-1 and 260108–008.107/2016).

## DNA Deposition

The following information was supplied regarding the deposition of DNA sequences:

DNA sequences are available at GenBank: MK887326, MK896864–MK896866, at BioProject using ID PRJNA540940, and at BioSample via accessions SAMN11490545–SAMN11490548.

## Data Availability

Data is available at Genbank under SRA numbers: SRR9006449, SRR9006450, SRR9006453, SRR9006454, SRR9006455, SRR9006456, SRR9006458, SRR9006459.

## Supplemental Information

Supplemental information for this article can be found online at http://dx.doi.org/10.7717/peerj.7543#supplemental-information.

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
