# Peer review of "Genome skimming is a low-cost and robust strategy to assemble complete mitochondrial genomes from ethanol preserved specimens in biodiversity studies"

_PeerJ, doi:10.7717/peerj.7543_

## Round 0.1 · original submission · Major Revisions

The two reviews are very thorough and I agree with both of them. I agree with Reviewer 1 in his/her point that the title may be a bit misleading: best to mention mitogenomics specifically in the title, otherwise some readers might not find in the paper what the title implies. Reviewer 2 notes that much has been done in this area before, and so it is essential that you cite the several papers indicated in that review. In truth, I think most of these comments can be accommodated relatively easily and that the decision might be "minor revisions", but I will maintain what the reviewers have suggested.

Reviewer 1 ·

Basic reporting

no comment

Experimental design

no comment

Validity of the findings

no comment

Additional comments

General comment:
The authors describe a method to efficiently sequence and annotate mitogenomes for non model taxa, using a variety of procedures and softwares. The pipeline they propose is an assemblage of known methods that are cleverly selected, and their methodology is clearly presented, however the authors’ claim for innovation is quite exaggerated and should be dropped as very close approaches have been used, including in similar groups, for a a couple of years.


Specific points:
- Title: “large scale data” is uninformative. Maybe “large scale molecular sequences” or something similar?

- The authors choose two examples, Rhinebothriidae and Streblidae (arguments for this “selection” are lacking, one can only suspect that these were the samples available at time of the study) but in any case do not treat them equally in many respects: discussion, literature etc… with the flies kept mostly on the side. Although references are (over?)abundant in general in the text, they become rare for Streblidae (see e.g. line 284, why not a couple of key references here?).

- The overall novelty of the proposed methods is certainly moot, see for example Brabec et al.(2016) IJP 46: 555-562 (for cestodes) among others who published very similar results for parasitic Platyhelminthes in the recent years. (Briscoe Par. Int. 2016, Digenea; Vanhove BMC Gen. 2018, Monogenea …).

- On the other hand the authors choose to mention e.g. Maldonado 2017 which is about genome not mtgenome.. The overall selection of literature should be filtered again with a better coverage and focus.
The claim of “taxon identification” (L78-80) is not correct if the taxon was not identified previously (name and reference sequence).

- Minimum coverage value should be included besides mean coverage value (L160-270 and table 2). This would allow for a better representation of the data quality.

- Also does the total number of sequences on L259 represent the number of sequences or of *mapped* sequences??

- The statement on line 294 is incorrect. See Egger et al. 2017 BMC Genomics for the presence of ATP8 in some flatworms! It would be correct for Neodermata though.

- The claim of cost effectiveness needs to be substantiated with figures. The authors do not present any information allowing, e.g. to estimate the cost of a complete mtgenome or a cost per specimen and comparative values.

- Bibliography: Capitals for authors names and genera in references are essential


Summary:
The paper is interesting but lacks much of its claimed novelty. It could be shortened and more focused, literature coverage should be completed, balance between the 2 selected test groups should be improved, the points listed above should be taken into account. Provided these remarks are considered, it is acceptable for publication, with low to medium priority.


Typos and details: (obviously redaction of the final part of the manuscript has been bit hurried …)
L95: remove second right bracket
L106: for evolutionary… (not “of”)
L126: them not the
L127: add “or” (or remove “either”)
L143: this should have been done beforehand, in any case accession numbers should be available in the paper
L160: replace “and “by “or”
L250: define abbreviation PE (even if looking obvious)
L255: Fig or Figure, be consistent
L333: cestodes
L349: transposition
L350: remove “the”
L352: support?, confirm?…. instead of follow
L355: remove double word
L356: add “in” before “acetabulate”
L357: one instead of in (?)
L368: choose “the” or “their” and complete in accordance
L373: add “for”before cestode
L375: can
L383: change “that”
L403: add “it requires” after i., for consistency
L405: “and is not” instead of “or are”
L451: revise citation form
L627-32: twice the same reference

Reviewer 2 ·

Basic reporting

No comments

Experimental design

No comments

Validity of the findings

No comments

Additional comments

This is an interesting paper dealing with mitogenome assembly based on second-generation sequencing from low amount of invertebrate DNA. The manuscript is well written, methods are sound and discussion and conclusion well supported by the data. Although those mitogenomes will certainly be very useful for researchers working on related taxa in invertebrate systematics, since the manuscript is presented mostly as a methodological contribution my main criticism is related to how novel the findings really are. The main contributions seem to be related to (1) use of NGS to obtain mitogenomes at low cost based on (2) low amount of DNA of museum specimens in genomics of (3) non-model organisms. Certainly those are exciting points brought by NGS, but they have been extensively explored in many recent papers. In addition, much of that literature has not been mentioned in the paper, what makes the manuscript perhaps look more novel that it really is.

A few examples:

https://www.taylorfrancis.com/books/e/9781498729161/chapters/10.1201%2F9781315120454-9
https://books.google.com.br/books?hl=pt-BR&lr=&id=cMItDwAAQBAJ&oi=fnd&pg=PT316&ots=vUVKbbRifL&sig=2DIPlqS0vLj2TXbuqs2B78lXhmY&redir_esc=y#v=onepage&q&f=false
https://www.pnas.org/content/early/2013/09/04/1314445110.short
https://journals.plos.org/plosone/article?id=10.1371/journal.pone.0138446
https://onlinelibrary.wiley.com/doi/full/10.1111/1755-0998.12466

So my main question: what is really novel in this manuscript besides the assembled mitogenomes? I understand that PeerJ focus on the on scientific validity and suitability to join the scholarly literature, but focusing on what is really novel will certainly make the paper more attractive to its intended audience.

A few minor points:

- Although I do agree that the biodiversity crisis is an important issue, it seems a bit disconnected to the rest of text. The intro could begin stronger dealing with topics that are closer to the paper main goals.
- Mitogenomes are very useful, but stating that "This set of attributes allows the mitogenome to be used for reliable taxon identification, which is the premise for any study involving biological organisms, specially in biodiversity." ignores all the issues that promoted the shift from mtDNA to nDNA in the last 15-20 years.
- In some points the text sounds a bit like coming from a thesis or dissertation, as in "Raw reads from each pair were pre-processed using a series of UNIX commands."
By the way, it would be fantastic to the readers to have those commands/scripts available as supplementary material, which does not seem to be the case in the current version.

---

## Round 0.2 · accepted · Accept

Thank you for addressing the reviewers' comments.